# Genetic components of human pain sensitivity: a protocol for a genome-wide association study of experimental pain in healthy volunteers

Annina B Schmid,[1] Kaustubh Adhikari,[2,3,4] Luis Miguel Ramirez-Aristeguieta,[5] Juan-Camilo Chacón-Duque,[2,6] Giovanni Poletti,[7] Carla Gallo,[7] Francisco Rothhammer,[8] Gabriel Bedoya,[9] Andres Ruiz-Linares,[10,11] David L Bennett[1]

ABS and KA contributed equally.

For numbered affiliations see end of article.

**Correspondence to**
Professor David L Bennett;
david.bennett@ndcn.ox.ac.uk

## ABSTRACT

**Introduction** Pain constitutes a major component of the global burden of diseases. Recent studies suggest a strong genetic contribution to pain susceptibility and severity. Whereas most of the available evidence relies on candidate gene association or linkage studies, research on the genetic basis of pain sensitivity using genome-wide association studies (GWAS) is still in its infancy. This protocol describes a proposed GWAS on genetic contributions to baseline pain sensitivity and nociceptive sensitisation in a sample of unrelated healthy individuals of mixed Latin American ancestry.

**Methods and analysis** A GWAS on genetic contributions to pain sensitivity in the naïve state and following nociceptive sensitisation will be conducted in unrelated healthy individuals of mixed ancestry. Mechanical and thermal pain sensitivity will be evaluated with a battery of quantitative sensory tests evaluating pain thresholds. In addition, variation in mechanical and thermal sensitisation following topical application of mustard oil to the skin will be evaluated.

**Ethics and dissemination** This study received ethical approval from the University College London research ethics committee (3352/001) and from the bioethics committee of the Odontology Faculty at the University of Antioquia (CONCEPTO 01–2013). Findings will be disseminated to commissioners, clinicians and service users via papers and presentations at international conferences.

## Strengths and limitations of this study

► We propose a genome-wide association study of both baseline pain sensitivity as well as nociceptive sensitisation.
► The study will be conducted in an admixed population in Colombia, with variable proportions of European, Native American and sub-Saharan African ancestry.
► Phenotypic data will be collected by a single trained examiner, thus producing high-quality measures.
► Because we are focusing on highly reproducible pain phenotypes obtained at a single centre, our initial sample size is limited to 1500–2000 participants. This will allow us to identify only genetic variants that have intermediate and large (but not small) effect sizes.

## INTRODUCTION

Pain constitutes a major component of the global burden of diseases, with lower back and neck pain representing the single leading cause for years lived with disability followed closely by migraine and other musculoskeletal disorders.[1] Pain is a multidimensional experience involving a highly complex interaction of physical, biochemical, physiological, cognitive, emotional, behavioural and sociocultural factors. Many studies have identified genetic factors in a range of chronic pain conditions.[2] Importantly, a growing number of studies in patient populations suggest that genetics is an important contributory factor to pain susceptibility and severity.[2–4] Of note, twin studies using either clinical pain outcomes or experimental pain models suggest that sensitivity to pain has a heritability of up to 55%.[5–10] Interestingly, the heritability varies greatly depending on the clinical pain outcome or the sensory modality tested in experimental pain models.[7] There is evidence that different sensory modalities may have distinct genetic components contributing to their variance in humans,[10] and this is consistent with animal models which underline the distinct neurobiology mediating different sensory modalities.[11]

A powerful technology for identifying genetic determinants of human complex phenotypes is genome-wide association studies (GWAS). However, GWAS analyses

of pain remain limited, mainly due to the high numbers of individuals required to enable adequate power, and the complexities of accurately phenotyping traits that ultimately represent a personal subjective perception.[2] In contrast to disease cohorts where pain variation may be influenced by the severity of the disease process or its treatment, experimental pain studies measuring baseline pain sensitivity have the advantage of studying one stimulus in a standard condition (eg, controlling intensity, location and stimulus duration). To date, several candidate gene studies have been performed to determine genetic influences on human pain sensitivity using experimental pain models,[5–9 12 13] however, to our knowledge, a full GWAS has not yet been reported. One genome-wide study evaluated association for single nucleotide polymorphisms (SNPs) revealed by exome sequencing in a subset of twins cohort that were identified as having particularly high or low heat pain sensitivity. Using pathway analysis, there was significant enrichment for variants in genes of the angiotensin pathway.[14] Whereas a direct link between experimental pain sensitivity and clinical pain severity is often not present,[15] there is some evidence that findings from experimental pain models can be predictive of clinically relevant pathological pain such as postoperative pain.[16] Irrespective of the association between experimental and pathological pain, understanding the genetic influences on experimental pain sensitivity will provide important biological insights into the mechanisms underlying pain sensitivity.

An important drawback of most genetic studies in the pain field so far is that they have been performed mostly in populations of European ancestry, thus, they have explored only a small fraction of human phenotypic and genetic diversity.[17 18] This is important in the study of pain as recent studies point to variable pain thresholds for European Americans, African Americans and Latinos.[19–21] However, it is not clear if this variation in pain thresholds relates to differences in neurobiological mechanisms or other factors such as social or cultural[22] parameters. In addition, recent studies suggest that a phenotype associated with increased sensitisation of the nociceptive system due to temporal summation may render people vulnerable to developing clinically relevant pain.[23] However, most studies focus on pain phenotyping in the naïve state rather than sensitised state; only a handful of studies have investigated a genetic component underlying nociceptive sensitisation.[7 24] We, therefore, intend to use an algogen (allyl isothiocyanate [AITC], an agonist of the ligand gated ion channel TRPA1) in order to sensitise the nociceptive system to replicate the changes that could occur in pathological pain states.

This protocol proposes a GWAS on genetic contributions to baseline pain sensitivity and nociceptive sensitisation in a sample of unrelated healthy individuals of mixed European/Native American/African ancestry. We will evaluate baseline cutaneous pain thresholds as well as the variation in sensitisation following mustard oil (AITC) application, a controlled model of tissue injury. We hypothesise that we will identify SNPs associated with experimental pain stimuli in the naïve and sensitised state. Elucidating the genetic basis of pain variation has the potential to reveal targets for future analgesic development. This can be translated into improved pain management potentially tailored to an individual's pain risk or resilience, including sensitivity differences between different human populations.

## METHODS AND ANALYSIS
This GWAS follows the Strengthening the Reporting of Genetic Association Studies guidelines.[25] A flow chart of the study procedure is detailed in figure 1.

### Participants
Healthy participants aged 18–40 will be recruited in Medellin, Colombia via public noticeboards at local Universities, distribution of flyers and through the local print media. In addition, we are inviting previous participants from the Consortium for the Analysis of the Diversity and Evolution of Latin America (CANDELA)[26] GWAS to participate in this project.

Recruiting healthy young participants has advantages in GWAS studies. They are less likely to have undetected illnesses or other problems that may influence their biological pathway of pain sensitivity. Young people will also have less overall accumulated exposure or risks from environmental (external) factors which may affect their pain sensitivity. Such factors increase the overall variability of participants' pain perception response and reduce the power of detecting genetic causes. Since most traits are affected by a combination of genetic and environmental factors, many studies including CANDELA tend to use young participants for genetic variant discovery.[27]

Participants will be excluded if they have chronic pain or any chronic medical condition (eg, diabetes, neurodegenerative, musculoskeletal or psychiatric conditions). Participants currently taking analgesics, anti-inflammatories, opioids, antihistamines, antidepressants or antiepileptic drugs will be excluded. Women who are pregnant or in their menstrual phase (self-report) will be excluded from the study. Participants will be advised to not smoke or consume coffee within 1 hour of testing, and to avoid psychoactive substances or alcohol within 8 hours prior to testing. Further exclusion criteria include current or past self-inflicted injuries, as well as dermatomal, traumatic or infectious conditions affecting the arm, and a history of severe allergic reactions to any kind of medication, materials, food or insect bites. Participants with moderate to severe anxiety (≥25 on the Hamilton Anxiety Rating Scale[28]) or severe depression (>15 on the 16-item Quick Inventory of Depressive Symptomatology Self-Reported (QIDS-SR16)[29] will be excluded from the study. Recruitment started in January 2013 and is predicted to take approximately 5–7 years.

### Procedure
Participants will attend a single appointment at the quantitative sensory testing (QST) laboratory at the Universidad

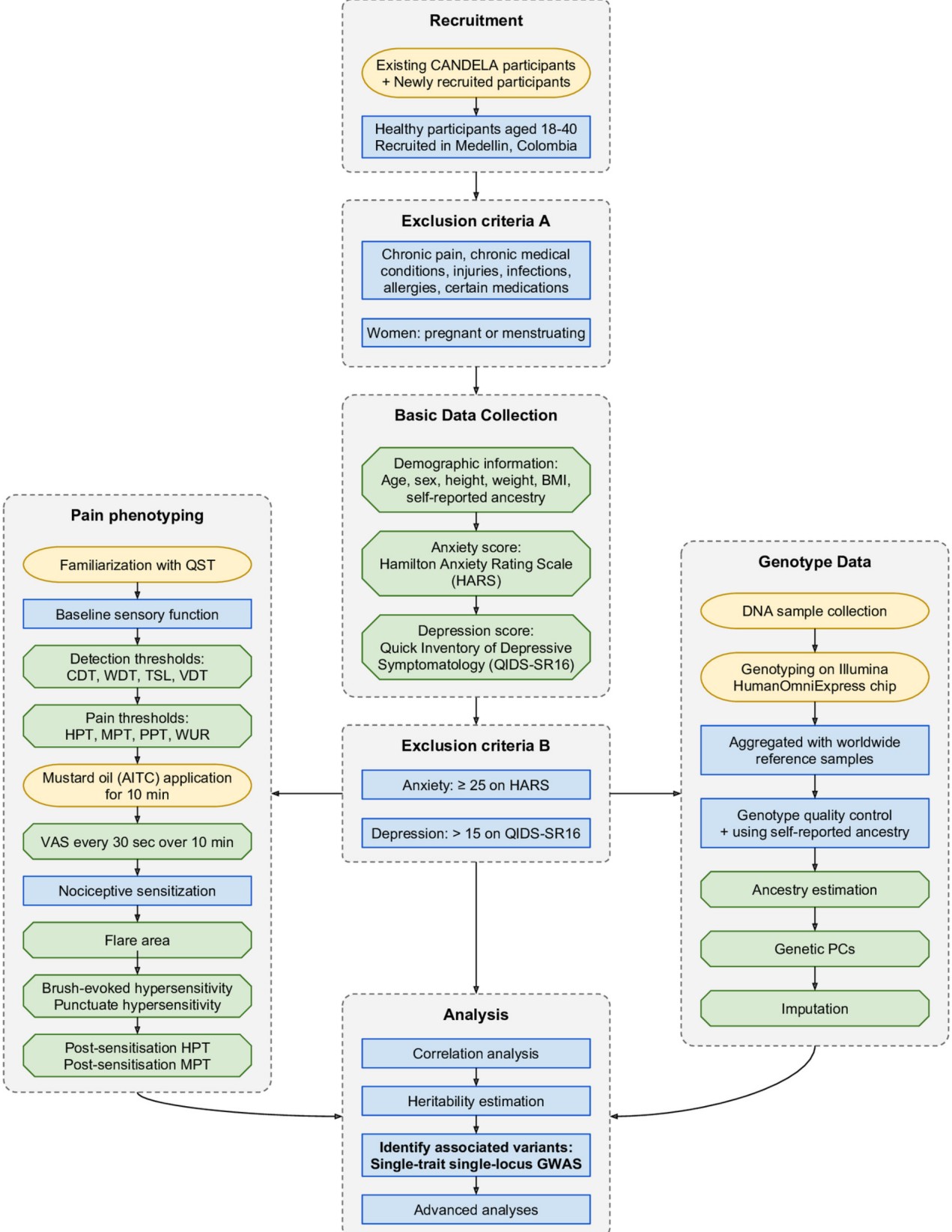

**Figure 1** Study procedure. The graph details the study procedure from recruitment to data analysis. Yellow boxes represent new procedures; green boxes indicate data generation and collection; blue boxes indicate a procedural step. AITC, allyl isothiocyanate; BMI, body mass index; CANDELA, Consortium for the Analysis of the Diversity and Evolution of Latin America; CDT, cold detection thresholds; GWAS, genome-wide association study; HPT, heat pain threshold; MPT, mechanical pain threshold; PCs, principal components; PPT, pressure pain threshold; QST, quantitative sensory testing; TSL, thermal sensory limen; VDT, vibration detection threshold; VAS, Visual Analogue Scale; WDT, warm detection thresholds; WUR, wind-up ratio.

de Antioquia, Medellín. Following informed consent, age and self-reported gender will be recorded and participants will answer questions regarding their self-reported ancestry (see online supplementary appendix 1). Height and weight will be measured and body mass index (BMI) calculated. Since psychological factors such as anxiety can influence pain perception during experimental pain testing,[30] participants will complete the Spanish version of the Hamilton Anxiety Rating Scale and the QIDS-SR16. The QIDS-SR16 has acceptable internal consistency and moderate to strong concurrent validity compared with other depression scores[31] and its Spanish version shows adequate test–retest reliability and high internal consistency.[32] The Hamilton Anxiety Rating Scale has shown to have high inter-rater and test–retest reliability[33] and good construct validity.[34]

### Evaluation of sensory function in the naïve state

We will determine sensory function in the naïve state and following nociceptive sensitisation. Baseline sensory function will be evaluated using specific static and dynamic QST. These include cold detection threshold and warm detection thresholds, thermal sensory limen and heat pain thresholds (HPT) using a ThermoTester (Q-sense, Medoc, Israel, 30×30 mm thermode size). Recording of thermal thresholds will strictly follow published QST guidelines.[35] Mechanical pain thresholds (MPT) will be evaluated using a 20 piece von Frey hair set (Touch Test, North Coast, USA) which exerts differing forces (9.8, 13.7, 19.6, 39.2, 58.8, 78.5, 98.1, 147.1, 255.0, 588.4, 980.7, 1765.2, 2942.0 mN). The von Frey hairs will be applied at a rate of 2 s on, 2 s off in ascending order starting from 9.8 mN baseline stimulus until participants first perceive the stimulus as sharp (pricking). Subsequently, the hairs will be applied in descending order until the stimulus is perceived as blunt. The geometric mean of five series of ascending and descending stimuli is defined as the MPT. Wind-up ratio will be determined with numerical pain ratings on a Visual Analogue Scale (VAS 0–100) for a single stimulus followed by the average pain rating for a train of 10 stimuli applied at 1 Hz within the same 1 cm$^2$ using a 255 mN von Frey hair. This will be repeated five times and the ratio will be established as the mean rating of the trains of stimuli divided by the mean rating of the single stimuli. Vibration detection thresholds (VDT) will be determined by recording the mean of 3 disappearance thresholds with a Rydel-Seiffer tuning fork. Pressure pain thresholds (PPT) will be recorded in triplicate with a manual algometer (Wagner Instruments, Greenwich, Connecticut, USA) and their mean used for analysis.

The side to be tested will be randomised and patients will first be familiarised with the sensory tests on the forearm on the control side, before performing the actual measurements on the test arm. All tests will be performed halfway over the volar side of the forearm except for VDT (ulnar styloid) and PPT (thenar muscles).

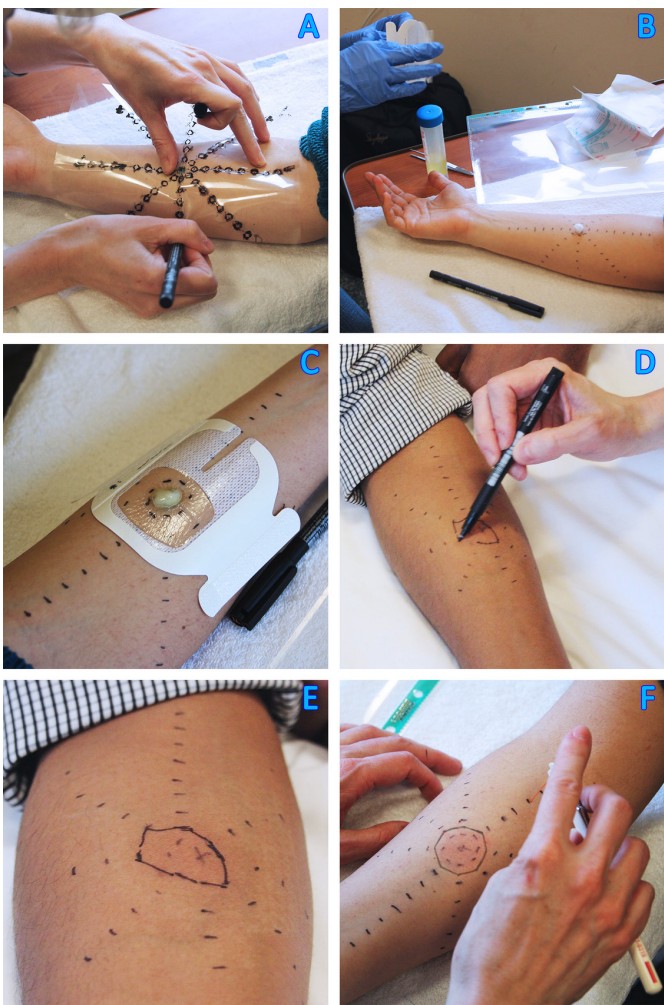

**Figure 2** Method to determine area of flare, punctuate hyperalgesia and allodynia. (A) An acetate template is used to mark a star with eight spokes containing eight points at 1 cm increments on the volar forearm. (B) A small cotton swab soaked in 30% mustard oil is applied in the center of the star and (C) held in place with a Tegaderm for 10 min. During this time, pain scores are recorded every 30 s. (D and E) After removal of the mustard oil, the skin flare will be marked and the area calculated. (F) The area of brush-evoked and punctuate hypersensitivity will be determined with a brush and a 98.1 mN von Frey hair respectively (pictured) by testing potential hypersensitivity at each point on the eight spokes.

### Mustard oil evoked nociceptive sensitisation

After the baseline sensory measures, an acetate template will be used to mark a star with eight spokes each containing eight points at 1 cm increments on the volar forearm (figure 2). The skin temperature will be standardised by placing the 32°C warm thermode over the center of the star for 5 min before starting. We will then apply a sensitisation paradigm using mustard oil (AITC (Sigma), diluted at 30% in olive oil) as previously performed.[36] AITC, the active component of mustard oil, activates the ion channel TRPA1 and evokes skin flare and nociceptive sensitisation.[37] A small cotton swab soaked in mustard oil will be applied to a 0.64 cm$^2$ area on the volar forearm and held in place with a Tegaderm (3M) for

10 min.[36] During this time, pain scores will be recorded every 30 s using an electronic VAS ranging from 0 to 100. After 10 min, the mustard oil will be removed and the area of the skin flare will be recorded to the nearest 0.5 cm at each spoke.[7] Eight triangular shapes will be created by joining the points on adjacent spokes and the total area will be calculated by adding all triangular segments. The area of mustard oil application will be subtracted from the total area to determine the area of secondary flare (flare area).

After mapping of the flare, the area of brush-evoked hypersensitivity will be determined with a brush (Nr 5 Senselab, Somedic, Sweden) by applying 1 cm long strokes at each of the points on the eight spokes, starting from the outside and moving towards the sensitised centre. The area of punctuate hypersensitivity will be determined with a 98.1 mN filament (Bailey Instruments, UK) following the same procedure.[36] As for the flare, the primary area of mustard oil application will be subtracted from both hypersensitive areas such that the recorded areas represent secondary hyperalgesia/hypersensitivity.

Following mustard oil sensitisation, the MPT and HPT will be repeated using the same methods as described above. All postsensitisation tests will be performed within 5 min of mustard oil removal.

### Reliability of naïve and sensitised sensory function protocol

To determine intratester reliability, we repeated the sensory function protocol performed by the same investigator in n=12 healthy volunteers on two different occasions within 2–6 weeks. Intraclass correlation coefficients (3.1) revealed good to excellent agreement for all sensory testing variables (table 1).[38]

**Table 1** Intratester reliability of sensory function protocol

| | Intraclass correlation coefficients | 95% CI | P value |
|---|---|---|---|
| CDT | 0.728 | 0.277 to 0.914 | 0.003 |
| WDT | 0.764 | 0.351 to 0.927 | <0.0001 |
| TSL | 0.638 | 0.161 to 0.878 | 0.005 |
| HPT | 0.752 | 0.339 to 0.922 | 0.002 |
| MPT | 0.928 | 0.767 to 0.979 | <0.0001 |
| WUR | 0.634 | 0.113 to 0.880 | 0.012 |
| VDT | 0.956 | 0.860 to 0.987 | <0.0001 |
| PPT | 0.734 | 0.305 to 0.915 | 0.002 |
| VAS | 0.893 | 0.667 to 0.970 | <0.0001 |
| Flare area | 0.610 | 0.095 to 0.869 | 0.015 |
| Brush-evoked allodynia | 0.756 | 0.365 to 0.922 | 0.001 |
| Punctuate hyperalgesia | 0.615 | 0.094 to 0.871 | 0.013 |
| Postsensitisation MPT | 0.941 | 0.808 to 0.983 | <0.0001 |
| Postsensitisation HPT | 0.758 | 0.339 to 0.924 | 0.002 |

CDT, cold detection threshold; HPT, heat pain threshold; MPT, mechanical pain threshold; PPT, pressure pain threshold; TSL, thermal sensory limen; VDT, vibration detection threshold; VAS, Visual Analogue Scale; WDT, warm detection threshold; WUR, wind-up ratio.

### Genotyping

Each participant will donate blood or saliva (Oragene OG-500, Genotek, Canada) for DNA extraction. DNA samples will be genotyped on the Illumina HumanOmniExpress chip containing ~700 000 markers. In volunteers who already participated in the CANDELA GWAS,[39] genotype data from blood samples genotyped on the same chip are already available and will be reused.

Whole-genome genotype data from the Illumina array will undergo quality control[40] to exclude any markers or samples that fail stringent thresholds. Quality metrics provided by the genotype calling algorithm in the Illumina GenomeStudio software,[41] such as the GenTrain score, cluster separation score, and excess heterozygosity rates will be used to filter poorly genotyped SNPs. Subsequent SNP-level and sample-level quality control thresholds such as missingness will be applied. Sex mismatch between records and genetic data of X and Y chromosomes will be checked. Only samples and SNPs that pass all criteria will be retained for analysis. Details of the currently used quality control protocol for CANDELA genotyped samples are provided in online supplementary appendix 2.

### Statistical analysis

#### Sample size calculation

The power for GWAS of experimental pain phenotypes for varying sample and effect sizes was estimated following the formulae described in Visscher et al.[42] Estimated power is shown for a range of effect sizes for experimental pain phenotypes taken from existing experimental pain studies. The statistical software R V.3.4.1[43] was used to perform the calculations and produce the figures. The codes are published on https://github.com/kaustubhad/gwas-power.

In whole-genome SNP-based GWAS studies, the association analysis is usually conducted with a multivariate linear regression model, where the trait values are regressed onto an SNP genotype (with additive coding) and other covariates which commonly include age, gender, BMI and genetic principal components (PCs). The p value threshold[39 42] for genome-wide significant associations is commonly $5 \times 10^{-8}$, while the threshold for a suggestive significant association is commonly $10^{-5}$. Formulae to calculate power in GWAS with genome-wide and suggestive significance thresholds are presented in online supplementary appendix 3, and power calculated for the current GWAS setting is shown in figure 3.

Figure 3A shows estimated power (in percentage) as a heatmap under the standard GWAS settings of using whole-genome genotyping data and a p value significance threshold of $5 \times 10^{-8}$. Sample size (n) varies from 100 to 5000, while the proportion of trait variance explained by the marker ($q^2$, in percentage) varies from 0.01% to 6%. As sample size increases, power increases quickly for a range of trait variance values to reach 100%.

Figure 3B shows estimated power (in percentage) under the same settings but a suggestive p value significance

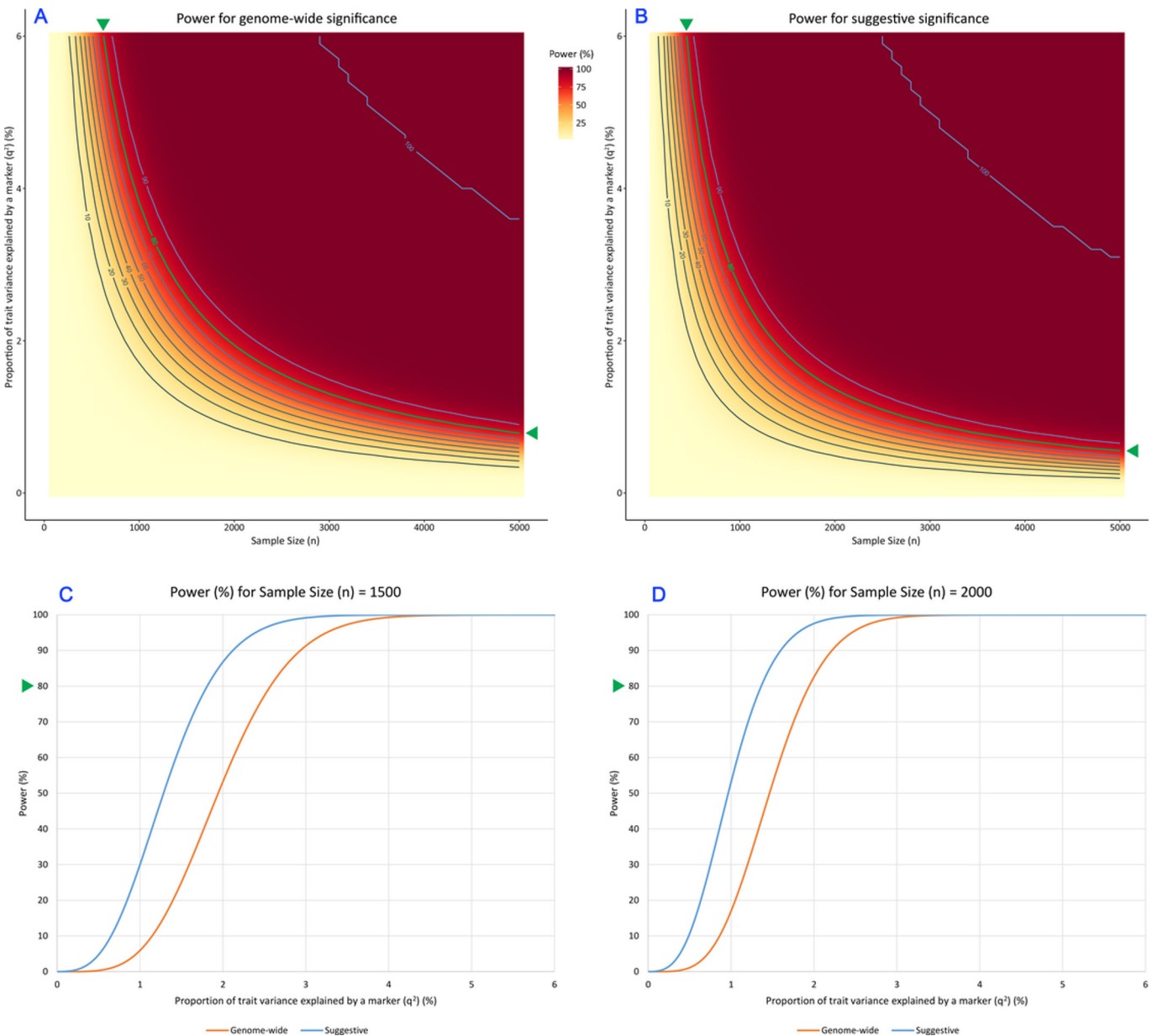

**Figure 3** Estimated power (in percentage) under the standard genome-wide association studies (GWAS) settings of using whole-genome genotyping data. (A) Estimated power (in percentage) as a heatmap, setting the significance threshold at $5\times10^{-8}$, the commonly used threshold for genome-wide significance in GWAS studies. (B) Estimated power with the significance threshold set at $10^{-5}$, the commonly used threshold for suggestive significance. In panels A-B, the x-axis denotes a range of sample sizes (n) in a GWAS, the y-axis represents the proportion of trait variance ($q^2$) explained by a marker. Power of detecting the marker at a specific (n, $q^2$) combination is represented by a colour gradient. Contour lines for power at 10% intervals are also shown. Panels C-D shows power curves for the expected sample sizes for this study. (C) Expected power at genome-wide and suggestive significance thresholds for a sample size of n=1500. (D) Estimated power for a sample size of n=2000. In Panels C-D, the x-axis denotes the proportion of trait variance ($q^2$) explained by a marker, and y-axis represents estimated power (in percentage). The two curves correspond to the two commonly used GWAS thresholds. In each panel, the point for 80% power is indicated with a green triangle, so that the necessary parameter configurations can be read from the graph. In panels A-B, the contour corresponding to 80% power is also marked in green.

threshold of $10^{-5}$. As expected, power is higher at similar sample and effect sizes for this less stringent threshold.

Simplified power estimates are shown as power curves in figure 3C,D for the expected sample sizes for this study. Figure 3C shows expected power at genome-wide and suggestive significance thresholds for a sample size of n=1500 at varying effect sizes, while figure 3D shows estimated power for a sample size of n=2000.

The range of trait variance has been taken from Doehring *et al*,[44] which provides estimates for the proportion of trait variance explained by an SNP for several experimental pain phenotypes and multiple markers. The

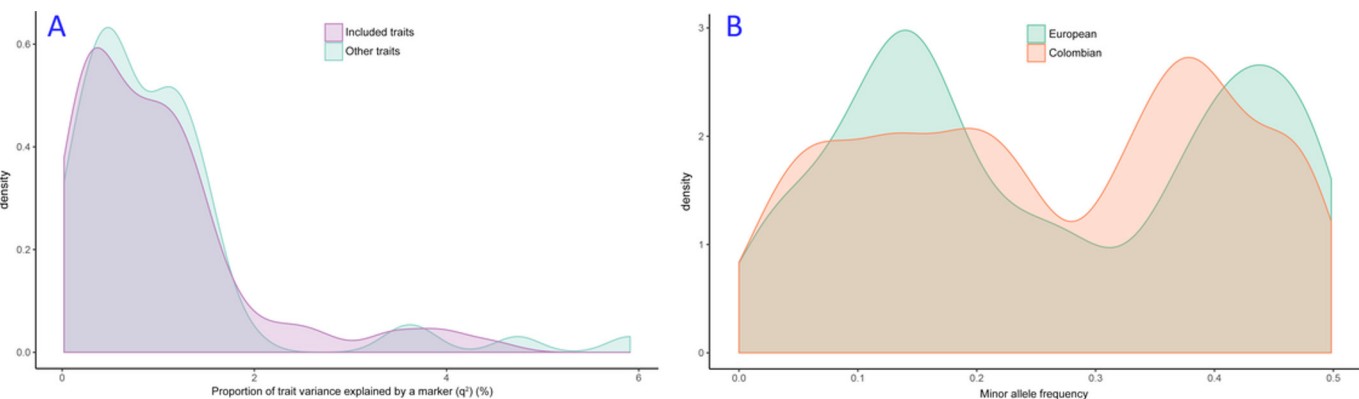

**Figure 4** (A) Distributions of trait variance explained by a single marker from Doehring *et al*[44] for traits included in our study and those not included. (B) Allele frequency distributions of loci associated with experimental pain in previously published cohorts, for Europeans and Colombians.

values ranged from 0.02% to 6%. Some of the traits were the same as the traits investigated here, while some other traits were different. Nevertheless, the distributions of trait variance for the two groups of traits are very similar, as seen in figure 4A.

Power of a GWAS depends on the allele frequency of the SNPs through their effect on the test statistic. While the majority of GWAS studies are conducted in European-origin individuals, including the experimental pain study used to determine sample size here,[44] our population of interest is an admixed Latin American population. Therefore, we wanted to assess the distribution of allele frequencies in Europeans versus Colombians for SNPs studied for or associated with experimental pain in various studies.[44 45] Minor allele frequencies were obtained for all such reported SNPs from the 1000 Genomes project database[17] for Western Europeans (from Britain (GBR), Utah residents from Northern and Western Europe (CEU), Spain (IBS), and Tuscany in Italy (TSI)) and Colombians (from Medellin in Colombia, (CLM), where this study will be performed). Allele frequency distributions for both Europeans and Colombians are shown in figure 4B. The two distributions are quite similar, with the Colombian distribution slightly more spread out. This is somewhat expected as the Colombians have on average 60% European (Spanish) ancestry.[26] Having a well spread out distribution of allele frequencies is important in a GWAS as low-frequency alleles have lower power for a given sample and effect size.[46] Here, the comparison to European allele frequency distribution suggests that the current Colombian cohort will have nearly equivalent power to any European-based cohort. In contrast to European-only cohorts though, our cohort will have the advantage that alleles present in other continental populations such as sub-Saharan Africans or Native Americans that are not present in Europeans would also be detectable in a GWAS, and could be followed up in replication cohorts of specific ethnicities.

The CANDELA project includes genotpyes of ~2000 patients from Medellin. We anticipate to contact and phenotype 50%–75% of these participants as well as

contacting an additional 500 participants to bring the initial sample to 1500–2000.

### Data analysis plan

The cleaned genetic data will first be merged with reference samples worldwide, such as the 1000 Genomes Project,[17] Simons Genome Diversity Project,[47] Estonian Biocentre Human Genome Diversity Panel,[48] and additional European and Native American samples that are particularly relevant for Latin American populations.[49] The merged dataset will be checked for genetic outliers, through genetic PCs and continental ancestry proportions (using supervised Admixture[50]), and for unexpected genetic similarities. These steps can often detect any sample misplacement or contamination, which might be reflected in sex mismatch, unexpected genetic similarities or inflated heterozygosity rates.[40 51] Genetic ancestry estimates will be compared with self-reported ancestry information (see online supplementary appendix 1), particularly for genetic outliers or samples showing unexpected results. Participants self-reporting for ethnicities rare in Colombians, such as East Asian or South Asian, would also be excluded as outliers. The authors have extensive experience in conducting association analysis in admixed populations, including several GWAS publications on a wide range of phenotypes which contain detailed protocols on how to conduct such analyses.[27 39 52 53] Further details of the currently used quality control protocol for CANDELA samples[53] are provided in online supplementary appendix 2.

The genetic data allow estimation of the narrow-sense heritability of any quantitative trait, which is the fraction of trait variance that is explained by the genetic data. Estimates of heritability, obtained using the software GCTA,[54] will provide an idea of which traits have more of a biological basis versus which are more environmentally determined, and thus which traits would be more amenable to genetic analysis for discovery of associated genetic variants. Note however that relatively precise estimate (low SDs) of heritability by this method requires several thousand samples,[52] so the currently proposed

sample size might be underpowered to estimate heritability accurately.

To facilitate better identification of associated loci, the genotype data will be imputed to approximately 10 million loci using the 1000 Genomes phase 3 imputation reference panel[52] by first haplotype phasing using SHAPEIT2[55] and then imputation using Impute2.[56] Quality control of the imputed genotypes will be performed using recommended thresholds on imputation quality score, concordance metrics and proportion of high-probability calls. Details of the currently used imputation protocol for CANDELA samples[52] are provided in online supplementary appendix 2.

GWAS studies will be conducted in Plink2[57] to perform single-locus association studies for each trait individually, across the whole genome in an additive multivariate linear regression model.[42] Covariates will be used in the regression to adjust for any other sources of trait variability, such as basic variables like age, sex, and BMI, and genetic PCs will be used to control for population substructure.[52 58]

The number of genetic PCs to be included in the regression depends on the sample composition, such as variation in ancestry and presence/absence of genetic outliers. It would be determined by inspecting the proportion of variance explained by each PC (displayed on a scree plot) and by checking PC scatter plots.

In addition to being used as exclusion criteria, anxiety and depression scores could be used as covariates in GWAS. The exact set of covariates to be used will be determined based on initial diagnostic analyses such as correlation analysis.

These single-locus association results, obtained as p values, will be visualised via the Manhattan plot. Commonly used p value thresholds for selecting associated loci are $5\times10^{-8}$ for genome-wide significance and $10^{-5}$ for suggestive significance.[42]

An extension of this additive multivariate linear regression model, still within the single-trait single-locus setting, called the mixed linear model analysis which better controls for any cryptic relatedness or population substructure, will also be performed in GCTA.[54]

There are several extensions of the single-trait single-locus association studies that increase power for detecting associated loci: combining several related traits that may share a biological basis, using multivariate Wald tests as implemented in MultiPhen[59]; or gene-based tests that combine signals across all loci in a gene to increase signal strength and reduce the burden of multiple testing, such as set-based models implemented in Plink2[57] or fastBAT implemented in GCTA.[60] The admixed nature of the sample might be used in detecting associations by the method of admixture mapping,[61] though the potential of success of this method in detecting associated variants depends on the extent of stratification of the variant's allele frequency across continents. These analyses might help detect additional loci that are underpowered in classical GWAS due to smaller effect sizes.

### Handling of missing data

It might not be possible to record some traits in some individuals, even though the completeness of the first 100 samples suggests that missingness will be low. The single-trait methods used in traditional GWAS analyses automatically exclude individuals from the analysis of a trait who have missing values for that trait. The same applies to individuals having missing genotypes for any particular SNP. However, genotyping success rate using the Illumina HumanOmniExpress chip in the CANDELA cohort is very high (>99.8%), so the number of excluded individuals in any analysis would be very low overall.

Some multivariate analyses such as PCs when applied on the set of phenotypes require having recorded values of all phenotypes for an individual. Instead of using the subset of individuals who have the complete set of phenotypes recorded, which would incur some loss in sample size, the missing phenotype data for each individual will be imputed following standard statistical procedures as implemented in the R package 'mice'.[62] When the proportion of missing data is small, imputation is preferable in such multivariate analyses than sample exclusion, and is routinely applied to genetic data such as while calculating genetic PCs.[58]

## DISCUSSION

This GWAS including a well-defined cohort of healthy participants will provide important insights into the genetic aspects underlying experimental pain sensitivity in the naïve and sensitised state. This may allow further exploration of potential biological mechanisms underlying pain sensitivity. Future studies will be required to extrapolate these findings to patient populations with chronic pain.

### Patient and public involvement

No patient involvement is performed during this study.

### Ethics and dissemination

Findings will be disseminated to commissioners, clinicians and service users via papers and presentations at international conferences such as the biennial World Congress of International Association for the Study of Pain. We will also post our findings to the publicly available painnetworks.org database.

**Author affiliations**
[1]Nuffield Department of Clinical Neurosciences, Oxford University, Oxford, UK
[2]Department of Genetics, Evolution and Environment, University College London, London, UK
[3]Department of Cell and Developmental Biology, University College London, London, UK
[4]School of Mathematics and Statistics, Faculty of Science, Technology, Engineering and Mathematics, The Open University, Milton Keynes, UK
[5]QST Lab. Faculty of Odontology, Universidad de Antioquia, Medellin, Colombia
[6]Department of Earth Sciences, Natural History Museum, London, UK
[7]Unidad de Neurobiologia Molecular y Genética, Laboratorios de Investigación y Desarrollo, Facultad de Ciencias y Filosofía, Universidad Peruana Cayetano Heredia, Lima, Peru
[8]Instituto de Alta Investigación, Universidad de Tarapacá, Arica, Chile

[9]GENMOL (Genética Molecular), Universidad de Antioquia, Medellin, Colombia
[10]Ministry of Education Key Laboratory of Contemporary Anthropology and Collaborative Innovation Center of Genetics and Development, School of Life Sciences and Human Phenome Institute, Fudan University, Shanghai, Shanghai, China
[11]CNRS, EFS, ADES, Aix-Marseille Université, Marseille, France

**Acknowledgements** We thank Peter Kamerman for his input to parts of the study design, and Macarena Fuentes-Guajardo for helping with the phenotyping photographs in figure 2. We would like to thank the CANDELA team for their contributions in organisation, recruitment, data collection, sample processing and genotyping: Alvaro Alvarado, Ana Carolina Orozco, Caio C. Silva de Cerqueira, Claudia Jaramillo, David Balding, Diana Rogel Diaz, Francisco de Ávila Becerril, Francisco M. Salzano, Francisco Quispealaya, Hugo Villamil-Ramírez, Ilich Jafet Moreno, Javier Mendoza-Revilla, Javier Rosique, Jorge Gómez-Valdés, Joyce De la Piedra, Lavinia Schuler-Faccini, Mónica Ballesteros Romero, Major Eugênio Correa de Souza Junior, Malena Hurtado, María Pizarro, María Teresa Del Solar, Marcelo Zagonel de Oliveira, Mari-Wyn Burley, Maria-Cátira Bortolini, Martha Granados Riveros, Miguel Ángel Contreras Sieck, Paola Everardo, Paola Everardo Martínez, Paola León-Mimila, Ricardo Cebrecos, Ricardo Gunski, Rodrigo Barquera Lozano, Rolando Gonzalez-José, Rosilene Paim, Ruth Rojas, Samuel Canizales-Quinteros, Sergeant João Felisberto Menezes Cavalheiro, Tábita Hunemeier, Valeria Villegas, Vanessa Granja, Vanessa Sarabia, Victor Acuña-Alonzo, Virginia Ramallo, Wendy Hart, William Arias and William Flores.

**Contributors** DLB and AR-L conceptualised the study. Funding was acquired by DLB, ABS, GB and AR-L. ABS, J-CC-D, KA and LMR-A have contributed to the study methodology. AR-L, KA, J-CC-D, GB, FR, CG and GP contributed to the design and execution of the CANDELA aspect of this study. The first draft of the protocol was prepared by ABS, DLB, KA and AR-L and all authors provided critical evaluation and approved the final manuscript.

**Funding** Work leading to this publication was funded by grants from: the Newton Fund Institutional Links Grant from the British Research Council (grant number 216398412), the Excellence Initiative of Aix-Marseille University— A*MIDEX (a French 'Investissements d'Avenir' programme) and from a Wellcome Trust Senior Scientist Fellowship to DLB (ref. no. 095698z/11/z and 202747/Z/16/Z). KA was supported by a Wellcome Investigator Award WT107055AIA (to C.D. Stern). ABS was supported by the National Institute for Health Research (NIHR) Biomedical Research Centre (BRC) Oxford.

**Disclaimer** The views expressed are those of the author(s) and not necessarily those of the NHS, the NIHR or the Department of Health.

**Competing interests** None declared.

**Patient consent for publication** Not required.

**Ethics approval** This study received ethical approval from the University College London research ethics committee (3352/001) and from the bioethics committee of the Odontology Faculty at the University of Antioquia (CONCEPTO 01–2013).

**Provenance and peer review** Not commissioned; externally peer reviewed.

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
