## [Reviewer comments · BMJ Open]

ARTICLE DETAILS

TITLE (PROVISIONAL)	Genetic components of human pain sensitivity: a protocol for a genome-wide association study of experimental pain in healthy volunteers
AUTHORS	Schmid, Annina B.; Adhikari, Kaustubh; Ramirez-Aristeguieta, Luis Miguel; Chacón-Duque, Juan-Camilo; Poletti, Giovanni; Gallo, Carla; Rothhammer, Francisco; Bedoya, Gabriel; Ruiz-Linares, Andres; Bennett, David

VERSION 1 - REVIEW

REVIEWER	Anne Estrup Olesen Department of Clinical Medicine, School of Medicine and Health, Aalborg University, Denmark
REVIEW RETURNED	09-Aug-2018

GENERAL COMMENTS	Genetic components of human pain sensibility: an experimental pain study The presented protocol describes a genome wide association study on genetic contributions to pain sensitivity in the naïve state and following nociceptive sensitization. The protocol is well written with an interesting and relevant aspect. However, some issues should be clarified before publication. Endpoints/outcomes: Primary and secondary endpoints should be specified in page 7 under aims. Also hypothesis should be included. A lot of outcomes are included in the protocol, but it is hard to see what is the main purpose. This should be clear throughout the protocol. I will recommend to include a short table where all endpoints are listed, and it is indicated which ones are primary and which ones are secondary. This will give an overview of all the included outcomes. Sample size: 1500-2000 participants is a lot to recruit. Thus, dates and period of recruitment (#5) should for sure be stated at page 7 "participants". I am not an expert in sample size calculation on genome-wide association studies. Thus, I cannot evaluate if 1500-2000 is a sufficient number, and the long description on page 11-16 "statistical analysis" is difficult to understand for people not in this area. Ethics and dissemination: (page 17) Is it corrects, that the protocol was approves by the bioethics committee of the Odontology Faculty at the University of Antioquia
---

	in 2013? It should be explained why approval was 5 years ago – and why study has not started yet. The reporting checklist for genetic association studies should be reviewed again, as I think some parts are missing even though a page number is indicated in the checklist. For example #5 where recruiting dates/periods should be provided in estimates to explain if it is a one year study or a 10 years study. Also #12c – it should be explained how missing data will be handled. What if one or more qst measures are missing for example?
--	--

REVIEWER	Blair H. Smith University of Dundee, Scotland I collaborate on an unrelated project with the senior author (David Bennett). No other interests to declare
REVIEW RETURNED	16-Oct-2018

GENERAL COMMENTS	This paper presents the methods for conducting a GWAS on experimental pain in a Colombian population of mixed ancestry. As the authors state, there is clear and growing evidence that pain is a genetically heritable trait, but there are limitations to our current knowledge through a relative lack of well-conducted studies in the field, addressing the full complexity of pain. One of these limitations has been a lack of high quality, well-standardised phenotyping, and this study sets out to address this. There have been some good GWAS studies on pain, based on questionnaire data (for example, those coming out from the UK Biobank), and these might be appropriate to refer to here, to place this study in context. This study focuses on experimentally evoked pain, before and after sensitisation, and this is novel for a study of this size, allowing the first such GWAS. The pain that will be evoked is likely to be accurately produced and recorded, allowing analysis of a very pure phenotype, much better than in most GWAS studies of pain to date, which rely on recalled/reported pain, likely to be more subjective. It is also important to conduct studies in populations other than European populations, and this is another novel aspect of this study. The study population is quite specific, and could result in certain bias. They will all be aged 18-40, recruited through advertisements. None of them will have pre-existing chronic pain or any other chronic illness. This means that the findings will not be applicable to many/most people who experience pain that requires long-term treatment, though they may shed light on some pain mechanisms. The effects of this sample selection should be discussed, including selection bias and the clinical implications of the findings. There is excellent detail on the procedures relating to clinical testing (QST and sensitisation). This includes assessment of intra-tester reliability of the sensitisation testing, which is reassuring for this non-standard procedure. However, the details on other phenotyping are less clear. “[C]linical data will be collected (e.g., age, gender, BMI, general health, blood pressure)” – rather than examples of the data to be collected, we should see the full list including how they will be collected (for example, how will general health be measured?). Similarly questions relating to ancestry need to be detailed and justified, rather than just
--

	described with vague examples. Two questionnaires will be administered (HARS and QIDS) – why these and how valid are they? Figure 1 betrays this focus in the study by not including non-physical aspects of data collection in the study procedure. The analysis plan is detailed with regard to genetics and statistics (perhaps more detailed than expected for the latter, but clearly described). This will be helpful when it comes to conducting and reporting their analysis. However, some detail on the determination of phenotypes to be tested for association with SNPs in the GWAS would be helpful. Will they simply test every available phenotype, conducting tens of separate genome-wide analyses, or look for clusters of phenotypes? Either or both approach could be appropriate, but better to specify in advance to avoid later accusations of multiple testing. And how will they use the clinical and questionnaire data in this process? In summary, therefore, this will be a novel and useful study, and I look forward to seeing the results. A little further explanation, as suggested above, would be helpful to read at this stage. Thank you for asking me to read the paper.
--	--

REVIEWER	Samar Khoury McGill University, Canada
REVIEW RETURNED	26-Nov-2018

GENERAL COMMENTS	In this protocol, the authors propose to do a GWAS on various QST measures. This proposition is novel and very important. As the authors stated, the field of pain genetics is still in its beginning in term of GWAS. This study, if completed, will be a great addition to the field. However, as the proposal is currently written, I am not sure that the authors understand the subtleties in performing analyses of this nature. I will enumerate below a series of anticipated difficulties that the authors failed to address The title: Do the authors mean sensitivity? My first major concern is the fact that the authors would like to infer vulnerability to clinical pain from QST measures. I understand that there are previous studies that showed an association between heat pain sensitivity and clinical pain, but there is a large body of literature showing that QST measures are not predictive for clinical pain. I suggest focusing on QST measures in this project proposal because a lot can be learned about the underlying genetic factors leading to pain perception. My second major concern is about the study population. The fact that the authors plan to do a GWAS in an admix population presents a major statistical challenge. As the authors themselves stated, previous GWASes were done in EUR ancestry populations for a reason. I would have liked to see how the authors plan to account for admixture. For example, will the author stratify? Will the authors do admixture mapping... Finally, the authors did not report any timeline for study completion. How long will it take to recruit? Are the proposed numbers of sample feasible in their setting? How long will it take to genotype?... Some minor points, why do you have two different means of collection (blood and saliva)? Were the CANDELA participants also genotyped on the HumanOmniExpress chip? If not, how are the authors planning to combine the two? In figure 3, is it possible to highlight 80% power as it will be more visible to the reader.
--

VERSION 1 – AUTHOR RESPONSE

Reviewer: 1

The presented protocol describes a genome wide association study on genetic contributions to pain sensitivity in the naïve state and following nociceptive sensitization. The protocol is well written with an interesting and relevant aspect. However, some issues should be clarified before publication.

R1.1 Endpoints/outcomes:

Primary and secondary endpoints should be specified in page 7 under aims. Also hypothesis should be included. A lot of outcomes are included in the protocol, but it is hard to see what is the main purpose. This should be clear throughout the protocol. I will recommend to include a short table where all endpoints are listed, and it is indicated which ones are primary and which ones are secondary. This will give an overview of all the included outcomes.

As this is a GWAS and not a classical clinical trial, we do not have distinct primary and secondary outcome measures. But as suggested by the reviewers, we have clarified our primary and secondary objectives in the manuscript. We now mention in our analysis plan that our primary analysis will be to identify genetic factors associated with single experimental pain stimuli. We have also provided a detailed flow chart of the recruitment including exclusion criteria, data collection, quality control and analysis procedures (Figure 1).

Also, following the reviewer's suggestion, the main hypothesis of variant identification in GWAS being the primary outcome has been highlighted in the introduction (page 7): 'We hypothesise that we will identify single nucleotide polymorphisms (SNPs) associated with experimental pain stimuli in the naïve and sensitized state.' We have also clarified in the methods and amended Figure 1 that we will conduct additional analyses.

Thank you for pointing out the ambiguity with the outcome measures. Most of the clinical data was used to screen for eligibility and we have therefore added these to the paragraph on exclusion criteria to make it clear that these were not outcome measures (see also Figure 1).

R1.2: Sample size: 1500-2000 participants is a lot to recruit. Thus, dates and period of recruitment (#5) should for sure be stated at page 7 "participants".

We have added the date of start of recruitment and clarified that the study is ongoing: Page 9: 'Recruitment started in January 2013 and is predicted to take approximately 5-7 years.'

R1.3: I am not an expert in sample size calculation on genome-wide association studies. Thus, I cannot evaluate if 1500-2000 is a sufficient number, and the long description on page 11-16 "statistical analysis" is difficult to understand for people not in this area.

The sample size calculation was performed in accordance with current practice in GWAS. As the readership will include clinicians as well as geneticists, we believe it is important to provide the full details to maintain transparency and enable similar analysis by other researchers. Following the reviewer's indication, however, we have moved the details of the calculation to Appendix 3, while keeping only the findings in the main text, so that a general reader would not be burdened with the details. Considering that interpreting the power heatmaps in Figure 3 can be difficult for some readers, we have also provided simplified power curves for the expected sample sizes of our study.

R1:4: Ethics and dissemination: (page 17)

Is it correct, that the protocol was approved by the bioethics committee of the Odontology Faculty at the University of Antioquia in 2013? It should be explained why approval was 5 years ago – and why study has not started yet.

Yes, this is correct and the study has started in 2013. We clarified this in the manuscript, please also see R1.2.

R1.5: The reporting checklist for genetic association studies should be reviewed again, as I think some parts are missing even though a page number is indicated in the checklist. For example #5 where recruiting dates/periods should be provided in estimates to explain if it is a one year study or a 10 years study.

We carefully reviewed the STREGA checklist. We have clarified a few points, however the STREGA checklist does not fully apply as our manuscript is a protocol of the GWAS rather than the results paper. Therefore, many STREGA questions especially the ones relating to results and interpretation of data are not applicable to this manuscript and therefore remain blank in the checklist. We have however clarified a few points suggested in STREGA:

Hypothesis Page 7: 'We hypothesise that we will identify single nucleotide polymorphisms (SNPs) associated with experimental pain stimuli in the naïve and sensitized state.'

Recruitment dates Page 9: 'Recruitment started in January 2013 and is predicted to take approximately 5-7 years.'

Details of outcome measures: Addition of Appendix 1 to detail data collection for demographic and ancestry data, Figure 1 to contain detailed study procedure.

R1.6: Also #12c – it should be explained how missing data will be handled. What if one or more qst measures are missing for example?

Thanks to the reviewer for this important suggestion. We have added a section in the manuscript to explain how missing data will be handled.

Reviewer: 2

R2.1: This paper presents the methods for conducting a GWAS on experimental pain in a Colombian population of mixed ancestry. As the authors state, there is clear and growing evidence that pain is a genetically heritable trait, but there are limitations to our current knowledge through a relative lack of well-conducted studies in the field, addressing the full complexity of pain. One of these limitations has been a lack of high quality, well-standardised phenotyping, and this study sets out to address this. There have been some good GWAS studies on pain, based on questionnaire data (for example, those coming out from the UK Biobank), and these might be appropriate to refer to here, to place this study in context.

There are indeed many studies suggesting genetic contributions to many chronic pain conditions. As suggested, we have added this in the introduction and cite a recent comprehensive review summarizing these findings:

Page 5: 'Many studies have identified genetic factors in a range of chronic pain conditions¹. Importantly, a growing number of studies in patient populations suggest that genetics is an important contributory factor to pain susceptibility and severity¹⁻³.'

R2.2: This study focuses on experimentally evoked pain, before and after sensitisation, and this is novel for a study of this size, allowing the first such GWAS. The pain that will be evoked is likely to be accurately produced and recorded, allowing analysis of a very pure phenotype, much better than in most GWAS studies of pain to date, which rely on recalled/reported pain, likely to be more subjective. It is also important to conduct studies in populations other than European populations, and this is another novel aspect of this study.

We thank you for your supportive comments. No changes required.

R2.3: The study population is quite specific, and could result in certain bias. They will all be aged 18-40, recruited through advertisements. None of them will have pre-existing chronic pain or any other chronic illness. This means that the findings will not be applicable to many/most people who experience pain that requires long-term treatment, though they may shed light on some pain mechanisms. The effects of this sample selection should be discussed, including selection bias and the clinical implications of the findings.

This study specifically includes a tightly defined cohort of healthy participants without a history of chronic pain as we are interested in the genetic contributions to naïve and sensitized sensitivity to experimental pain in a highly controlled setting. We agree that our study does not allow direct conclusions to patients in chronic pain. We hope however that our findings will facilitate further work into the biological mechanisms and targets involved in pain sensitivity, as the reviewer correctly points out. Eventually, the newly acquired knowledge will have to be translated to patients with chronic pain conditions.

We have clarified the anticipated output of this study at the end of the manuscript under the heading 'Discussion':

'This GWAS including a well-defined cohort of healthy participants will provide important insights into the genetic aspects underlying experimental pain sensitivity in the naïve and sensitized state. This may allow further exploration of potential biological mechanisms underlying pain sensitivity. Future studies will be required to extrapolate these findings to patient populations with chronic pain.'

We have also clarified in the manuscript the advantages of using healthy young participants in GWAS studies (page 8-9, participants). For traits influenced by both genetic and environmental factors, such participant cohorts are particularly useful. For example, as we cite in the manuscript, a recent publication from the CANDELA cohort⁴ discovered one of the first genes associated with grey hair. Grey hair is common with aging due to environmentally accumulated stress, so by looking at early aging in young participants the study was successful in discovering associated genetic variants.

R2.4: There is excellent detail on the procedures relating to clinical testing (QST and sensitisation). This includes assessment of intra-tester reliability of the sensitisation testing, which is reassuring for this non-standard procedure. However, the details on other phenotyping are less clear. "[C]linical data will be collected (e.g., age, gender, BMI, general health, blood pressure)" – rather than examples of the data to be collected, we should see the full list including how they will be collected (for example, how will general health be measured?).

As explained under R1.1 above, most of the clinical data recorded was in fact to check for study eligibility rather than outcome measures. We have clarified this in the text by detailing the exclusion criteria as well as the clinical data recorded as clinical outcome measures or covariates in Appendix 1, and Figure 1.

R2.5: Similarly questions relating to ancestry need to be detailed and justified, rather than just described with vague examples.

We have added the details of the ancestry questions as Appendix 1. These questions were chosen as they were also part of the CANDELA cohort, a subset of which will also be used in our analyses. Their use is now clarified in the manuscript.

R2.6: Two questionnaires will be administered (HARS and QIDS) – why these and how valid are they?

We have clarified the use and psychometric properties of the HARS and QIDS in the manuscript, page 9:

'Since psychological factors such as anxiety can influence pain perception during experimental pain testing⁵, participants will complete the Spanish versions of the Hamilton Anxiety Rating Scale⁶ and the Quick Inventory of Depressive Symptomatology (QIDS-SR16)⁷. The QIDS-SR16 has acceptable internal consistency and moderate to strong concurrent validity compared to other depression scores⁸ and its Spanish version shows adequate test-retest reliability and high internal consistency⁹. The Hamilton Anxiety Rating scale has shown to have high inter-rater and test-retest reliability¹⁰ and good construct validity¹¹.'

R2.7: Figure 1 betrays this focus in the study by not including nonphysical aspects of data collection in the study procedure.

We have extended Figure 1 to a detailed protocol of the data collection, quality control and analysis procedures presented as a flowchart.

R2.8: The analysis plan is detailed with regard to genetics and statistics (perhaps more detailed than expected for the latter, but clearly described). This will be helpful when it comes to conducting and reporting their analysis. However, some detail on the determination of phenotypes to be tested for association with SNPs in the GWAS would be helpful. Will they simply test every available phenotype, conducting tens of separate genome-wide analyses, or look for clusters of phenotypes? Either or both approaches could be appropriate, but better to specify in advance to avoid later accusations of multiple testing.

We thank the reviewer for this point, as we agree it is important to provide this clarification at the protocol stage. The reviewer is correct that the main analysis will be single-trait single-SNP classical GWAS, which we clarify in the manuscript now, and highlight in the protocol Figure 1. We mention that more advanced analysis such as multivariate associations will subsequently be performed as additional analysis.

R2.9: And how will they use the clinical and questionnaire data in this process?

The Hamilton scale and QIDS-SR16 will be used in two ways: 1) participants with severe depression/anxiety according to these scales will be excluded. 2) these questionnaires will potentially be used as covariates in the analysis as it is well established that pain sensitivity is influenced by psychological factors such as anxiety. We have further clarified this in the methods section and Figure 1.

R2.10: In summary, therefore, this will be a novel and useful study, and I look forward to seeing the results. A little further explanation, as suggested above, would be helpful to read at this stage. Thank you for asking me to read the paper.

We thank you for your valuable input.

Reviewer: 3

R3.1: In this protocol, the authors propose to do a GWAS on various QST measures. This proposition is novel and very important. As the authors stated, the field of pain genetics is still in its beginning in term of GWAS. This study, if completed, will be a great addition to the field. However, as the proposal is currently written, I am not sure that the authors understand the subtleties in performing analyses of this nature. I will enumerate below a series of anticipated difficulties that the authors failed to address

Thank you for your valuable comments. Our team combines in-depth experience in sensory phenotyping of large cohorts^{12 13} as well as in large-scale genetic analyses required for this project (references in R3.4 below). We have carefully answered the queries of Reviewer 3 below.

R3.2: The title: Do the authors mean sensitivity?

We have adjusted the title to include 'sensitivity'.

R3.3: My first major concern is the fact that the authors would like to infer vulnerability to clinical pain from QST measures. I understand that there are previous studies that showed an association between heat pain sensitivity and clinical pain, but there is a large body of literature showing that QST measures are not predictive for clinical pain. I suggest focusing on QST measures in this project proposal because a lot can be learned about the underlying genetic factors leading to pain perception.

We agree with the reviewer that irrespective of a link between experimental and clinical pain, understanding the genetics with pain sensitivity will provide valuable knowledge. We have therefore moderated our statement:

'Whereas a direct link between experimental pain sensitivity and clinical pain severity is often not present¹⁴, there is some evidence that findings from experimental pain models can be predictive of clinically relevant pathological pain such as post-operative pain¹⁵. Irrespective of the association between experimental and pathological pain, understanding the genetic influences on experimental pain sensitivity will provide important biological insights into the mechanisms underlying pain sensitivity.'

R3.4: My second major concern is about the study population. The fact that the authors plan to do a GWAS in an admix population presents a major statistical challenge. As the authors themselves stated, previous GWASes were done in EUR ancestry populations for a reason. I would have liked to see how the authors plan to account for admixture. For example, will the author stratify? Will the authors do admixture mapping...

We explain in our manuscript, and reviewer 2 concurs in point R2.2, that using an admixed population is an advantage in identifying novel associated variants. We also agree with reviewer 3 that analysis in an admixed cohort is statistically more challenging than a homogenous cohort. The current authors however are experts in conducting association analysis in admixed populations, successfully conducting several GWAS publications on a wide range of phenotypes^{16 4 17} which contain detailed protocols on how to conduct such analyses, as well as a review paper¹⁸. The common strategy of controlling for population substructure or heterogeneity in conducting classical GWAS in admixed cohorts is to perform extensive genetic quality control and subsequently use genetic PCs as covariates, as we clarify in the manuscript. The reviewer is also correct in mentioning admixture mapping, which we will use as additional analysis. It is now mentioned in the manuscript.

R3.5: Finally, the authors did not report any timeline for study completion. How long will it take to recruit? Are the proposed numbers of sample feasible in their setting? How long will it take to genotype?

We started recruitment in January 2013. Given the preliminary recruitment rates, we anticipate that data collection for the 1500-2000 participants will take between 5-7 years. We have specified this in the manuscript page 7: 'Recruitment started in January 2013 and is predicted to take approximately 5-7 years.'

Given the access to state-of-the-art facilities for DNA extraction and genotyping (e.g., Wellcome Trust Centre for Human Genetics in Oxford), genotyping will be performed in batches of ~400 samples which takes about 3 months for sample preparation and processing.

R3.6: Some minor points, why do you have two different means of collection (blood and saliva)?

The original CANDELA participants were genotyped on blood samples. As whole blood cannot be imported from Colombia to the UK, and access to DNA extraction is limited in Medellin, we also use saliva collection tubes, which can be stored at room temperature and do not need additional processing before their import to the UK.

R3.7: Were the CANDELA participants also genotyped on the HumanOmniExpress chip? If not, how are the authors planning to combine the two?

CANDELA participants were also genotyped on the HumanOmniExpress chip, facilitating the combination of the cohorts. This is now clarified in the manuscript. Thanks for raising this important question.

R3.8: In figure 3, is it possible to highlight 80% power as it will be more visible to the reader.

We have highlighted 80% power in Figure 3.

References

1. Zorina-Lichtenwalter K, Meloto CB, Khoury S, et al. Genetic predictors of human chronic pain conditions. *Neuroscience* 2016;338:36-62. doi: 10.1016/j.neuroscience.2016.04.041
2. Bennett DL, Woods CG. Painful and painless channelopathies. *Lancet Neurol* 2014;13(6):587-99. doi: 10.1016/S1474-4422(14)70024-9
3. Denk F, McMahon SB. Neurobiological basis for pain vulnerability: why me? *Pain* 2017;158 Suppl 1:S108-S14. doi: 10.1097/j.pain.0000000000000858
4. Adhikari K, Fontanil T, Cal S, et al. A genome-wide association scan in admixed Latin Americans identifies loci influencing facial and scalp hair features. *Nature communications* 2016;7:10815. doi: 10.1038/ncomms10815 [published Online First: 2016/03/02]
5. Ignatiadis M, Polyzos A, Stathopoulos GP, et al. A multicenter phase II study of docetaxel in combination with gefitinib in gemcitabine-pretreated patients with advanced/metastatic pancreatic cancer. *Oncology* 2006;71(3-4):159-63. doi: 10.1159/000106064 [published Online First: 2007/07/25]
6. Hamilton M. The assessment of anxiety states by rating. *Br J Med Psychol* 1959;32(1):50-5.
7. Rush AJ, Trivedi MH, Ibrahim HM, et al. The 16-Item Quick Inventory of Depressive Symptomatology (QIDS), clinician rating (QIDS-C), and self-report (QIDS-SR): a psychometric evaluation in patients with chronic major depression. *Biol Psychiatry* 2003;54(5):573-83.
8. Reilly TJ, MacGillivray SA, Reid IC, et al. Psychometric properties of the 16-item Quick Inventory of Depressive Symptomatology: a systematic review and meta-analysis. *J Psychiatr Res* 2015;60:132-40. doi: 10.1016/j.jpsychires.2014.09.008 [published Online First: 2014/10/11]
9. Gili M, Lopez-Navarro E, Homar C, et al. Psychometric properties of Spanish version of QIDS-SR16 in depressive patients. *Actas Esp Psiquiatr* 2014;42(6):292-9. [published Online First: 2014/11/13]
10. Shear MK, Vander Bilt J, Rucci P, et al. Reliability and validity of a structured interview guide for the Hamilton Anxiety Rating Scale (SIGH-A). *Depress Anxiety* 2001;13(4):166-78. [published Online First: 2001/06/20]

11. Clark DB, Donovan JE. Reliability and validity of the Hamilton Anxiety Rating Scale in an adolescent sample. *J Am Acad Child Adolesc Psychiatry* 1994;33(3):354-60. doi: 10.1097/00004583-199403000-00009 [published Online First: 1994/03/01]
12. Themistocleous AC, Ramirez JD, Shillo PR, et al. The Pain in Neuropathy Study (PiNS): a cross-sectional observational study determining the somatosensory phenotype of painful and painless diabetic neuropathy. *Pain* 2016;157(5):1132-45. doi: 10.1097/j.pain.0000000000000491
13. Schmid AB, Bland JD, Bhat MA, et al. The relationship of nerve fibre pathology to sensory function in entrapment neuropathy. *Brain* 2014;137(Pt 12):3186-99. doi: 10.1093/brain/awu288
14. Hubscher M, Moloney N, Leaver A, et al. Relationship between quantitative sensory testing and pain or disability in people with spinal pain-a systematic review and meta-analysis. *Pain* 2013;154(9):1497-504. doi: 10.1016/j.pain.2013.05.031 [published Online First: 2013/05/29]
15. Werner MU, Mjobo HN, Nielsen PR, et al. Prediction of postoperative pain: a systematic review of predictive experimental pain studies. *Anesthesiology* 2010;112(6):1494-502. doi: 10.1097/ALN.0b013e3181dcd5a0
16. Adhikari K, Fuentes-Guajardo M, Quinto-Sanchez M, et al. A genome-wide association scan implicates DCHS2, RUNX2, GLI3, PAX1 and EDAR in human facial variation. *Nature communications* 2016;7:11616. doi: 10.1038/ncomms11616 [published Online First: 2016/05/20]
17. Adhikari K, Reales G, Smith AJ, et al. A genome-wide association study identifies multiple loci for variation in human ear morphology. *Nature communications* 2015;6:7500. doi: 10.1038/ncomms8500 [published Online First: 2015/06/25]
18. Adhikari K, Mendoza-Revilla J, Chacon-Duque JC, et al. Admixture in Latin America. *Curr Opin Genet Dev* 2016;41:106-14. doi: 10.1016/j.gde.2016.09.003 [published Online First: 2016/10/01]

VERSION 2 – REVIEW

REVIEWER	Anne Estrup Olesen Aalborg University, Aalborg, Denmark
REVIEW RETURNED	21-Jan-2019

GENERAL COMMENTS	I have no further comments - the manuscript has been improved sufficiently.
---

REVIEWER	Blair H. Smith University of Dundee, Scotland I collaborate with the senior author, in a current unrelated project.
REVIEW RETURNED	18-Jan-2019

GENERAL COMMENTS	The authors have addressed all previous concerns comprehensively and clearly. Thank you
---

REVIEWER	Samar Khoury McGill University, Canada
REVIEW RETURNED	30-Jan-2019

GENERAL COMMENTS	I have reviewed the re-submitted manuscript (bmjopen-2018-025530) and I have no more revisions to suggest. My recommendation is to accept.
--